# External validation of the PAR-Risk Score to assess potentially avoidable hospital readmission risk in internal medicine patients

Lukas Higi[1,2☯], Angela Lisibach[3,4,5☯], Patrick E. Beeler[6], Monika Lutters[3], Anne-Laure Blanc[7], Andrea M. Burden[8], Dominik Stämpfli[3,8]*

1 Department of Pharmaceutical Sciences, University of Basel, Basel, Switzerland, 2 PEDeus Ltd., Zurich, Switzerland, 3 Department Medical Services, Clinical Pharmacy, Cantonal Hospital of Baden, Baden, Switzerland, 4 School of Pharmaceutical Sciences, University of Geneva, Geneva, Switzerland, 5 Institute of Pharmaceutical Sciences of Western Switzerland, University of Geneva, University of Lausanne, Lausanne, Switzerland, 6 Division of Occupational and Environmental Medicine, Epidemiology, Biostatistics and Prevention Institute, University of Zurich and University Hospital Zurich, Zurich, Switzerland, 7 Clinical Pharmacy, Pharmacy of Eastern Vaud Hospitals, Rennaz, Switzerland, 8 Department of Chemistry and Applied Biosciences, Institue of Pharmaceutical Sciences, Swiss Federal Institute of Technology, Zurich, Switzerland

☯ These authors contributed equally to this work.
* dominik.staempfli@pharma.ethz.ch

## Abstract

### Background

Readmission prediction models have been developed and validated for targeted in-hospital preventive interventions. We aimed to externally validate the Potentially Avoidable Read-mission-Risk Score (PAR-Risk Score), a 12-items prediction model for internal medicine patients with a convenient scoring system, for our local patient cohort.

### Methods

A cohort study using electronic health record data from the internal medicine ward of a Swiss tertiary teaching hospital was conducted. The individual PAR-Risk Score values were calculated for each patient. Univariable logistic regression was used to predict potentially avoidable readmissions (PARs), as identified by the SQLape algorithm. For additional analyses, patients were stratified into *low*, *medium*, and *high* risk according to tertiles based on the PAR-Risk Score. Statistical associations between predictor variables and PAR as outcome were assessed using both univariable and multivariable logistic regression.

### Results

The final dataset consisted of 5,985 patients. Of these, 340 patients (5.7%) experienced a PAR. The overall PAR-Risk Score showed rather poor discriminatory power (C statistic 0.605, 95%-CI 0.575–0.635). When using stratified groups (*low*, *medium*, *high*), patients in the *high*-risk group were at statistically significant higher odds (OR 2.63, 95%-CI 1.33–5.18) of being readmitted within 30 days compared to *low* risk patients. Multivariable logistic

**Funding:** The authors received no specific funding for this work.

**Competing interests:** The authors have declared that no competing interests exist.

regression identified previous admission within six months, anaemia, heart failure, and opioids to be significantly associated with PAR in this patient cohort.

## Conclusion

This external validation showed a limited overall performance of the PAR-Risk Score, although higher scores were associated with an increased risk for PAR and patients in the *high*-risk group were at significantly higher odds of being readmitted within 30 days. This study highlights the importance of externally validating prediction models.

## Introduction

Potentially avoidable readmissions (PAR) are unforeseen readmissions related to a previously known affliction occurring within a specified time interval [1]. PAR-rates vary between 5 to 79% and are increasingly being used as benchmarks for quality of care, hospital outcomes, and cost reduction measures [2,3]. For Switzerland, the rates vary between 3.8% to 5.6% and generally depend on the level of care the hospital provides [4]. Of the many reasons for hospital readmissions investigated, adverse drug events have been shown to account for 13% of 30-day readmissions to an academic hospital in the US. Of these, 93% were classified as preventable and 49% were caused by inappropriate prescribing [5,6]. Interventions to reduce readmission rates have been explored by focussing on improved discharge planning and reducing adverse drug events, including patient education, telephone follow-up, home visits, and transition coaching [5]. However, no multicomponent intervention program has yet brought consistent evidence to sufficiently reduce readmission rates [5]. Furthermore, they are reported to be time consuming and expensive [7].

To address these issues, readmission prediction models have been developed and validated [7]. These models stratify patients according to their risk of readmission using readily available electronic health care information to calculate a risk score early in the hospitalisation in order to target interventions [8,9]. A recent systematic review on prediction models by Mahmoudi *et al.* identified 41 studies reporting on prediction models. Of these, 17 predict the risk of readmission on all inpatients while the rest of the models focus on a specific patient group [7]. Only eight studies reported sensitivity and specificity, implementation in the electronic medical record system seemed rare, and no model had been externally validated [10].

In Switzerland, the national Striving for Quality Level and Analyzing of Patient Expenses (SQLape) software [11] is being used to identify PAR within 30 days after discharge. The underlying screening algorithm identifies unplanned readmissions to the same hospital that are related to the initial diagnosis and occur within 30 days of hospital discharge with a specificity and sensitivity of 96% [1]. However, the screening algorithm of SQLape only works in retrospect and, hence, cannot be used for targeted preventive interventions. In 2019, Blanc *et al.* [12] published the internally validated Potentially Avoidable Readmissions Risk Score (PAR-Risk Score), developed with a dataset of one tertiary university teaching hospital and one regional hospital. The PAR-Risk Score is a 12-items prediction model, where the weights of the regression coefficients were transposed to a simpler scoring system. The internal validation showed a C-statistic of 0.688 (95% CI 0.655 to 0.72), which is close to the models reviewed by Mahmoudi *et al.* [7] and the already externally validated Swiss HOSPITAL score [13,14].

In this study, we aimed to provide the first external validation of the PAR-Risk Score using data from an older internal medicine patient cohort of a Swiss tertiary teaching hospital.

## Methods

### Study design and participants

This cohort study used data from hospitalisations of a 360-bed tertiary teaching hospital in Baden, Switzerland [15] between December 2016 and November 2018. As in the development study, we focused on internal medicine patients. The data were derived from routinely collected electronic health records (EHR). The study included patients hospitalised for at least 48 hours and aged 65 years and older, as selected for another study [16]. These dataset characteristics had not been applied to the patient sample in the development study. We applied the exclusion criteria according to the development phase of the model [12], namely: death before discharge, transfer to another hospital, and non-Swiss residents. The study design is visualised in Fig 1.

We report this study in concordance with the Transparent Reporting of a multivariable prediction model for Individual Prognosis Or Diagnosis (TRIPOD) statement [17].

### Outcome

The outcome of interest was a 30-day potentially avoidable hospital readmission (PAR), as identified by the SQLape algorithm.

### Predictors

The PAR Risk Score assigns points to the following predictors: length of stay longer than four days, admission in previous six months, anaemia, hypertension, hyperkalaemia, opioid prescription during hospital stay, comorbidities such as heart failure, acute myocardial infarction, chronic ischemic heart disease, diabetes with organ damage, cancer, and metastatic carcinoma. The exact scoring system with the individual weights is presented in S1 Table. Comorbidities were defined using International Classification of Disease 10 (ICD-

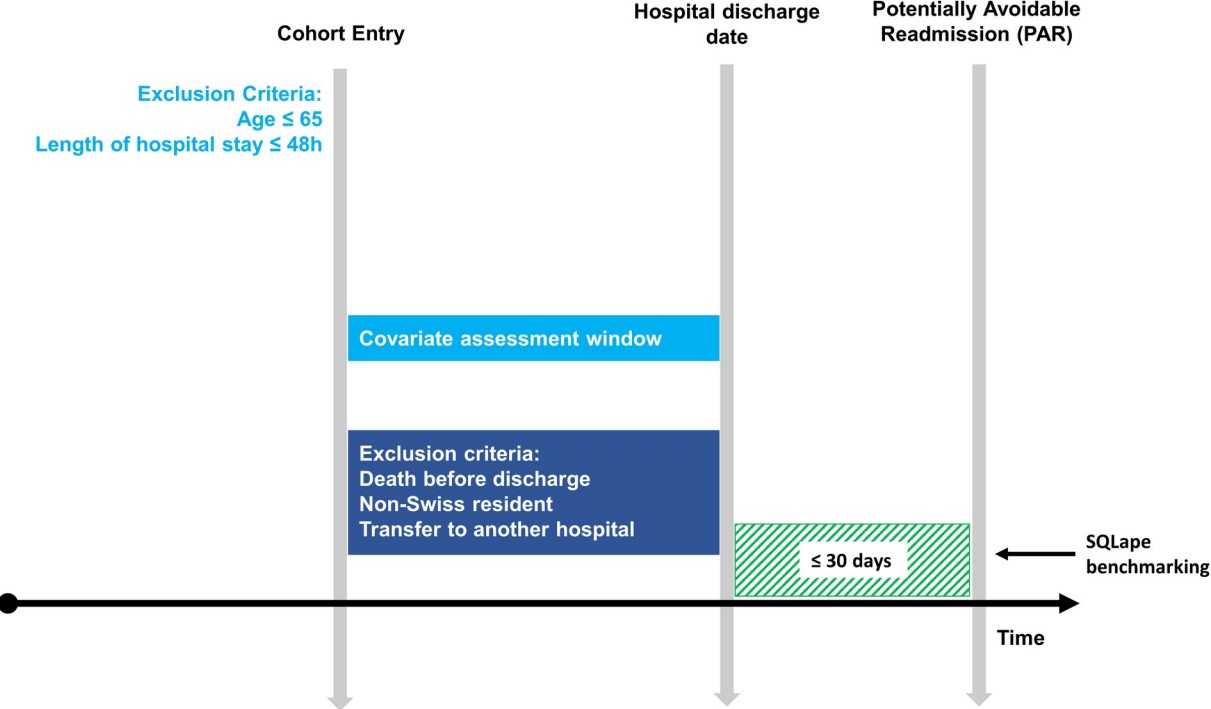

**Fig 1. Cohort design of the external validation of the PAR Risk Score.**

10) codes, as extracted from the EHR, and categorised according to the supplement information of the original publication [12]. For the comorbidity predictor anaemia only ICD-10 codes were used, because haemoglobin values were not available in this dataset. Opioid intake was defined as opioid drugs dispensed during the period of hospitalisation, as identified by the ATC-code *N02A* and its sub-levels. Medication prescription data was mapped in a semi-automated process to identify non-standardized free-text entries as well. Hyperkalaemia was defined as serum potassium level of >5.5 mmol/L within the last seven days of the hospitalisation.

## Missing data

Within individual patients, the most recent serum potassium levels were carried forward. When there was no value available at all, we assumed that those were not missing at random (i.e., the patient had normal serum potassium levels). A sensitivity analysis was performed by setting the hyperkalaemia variable for all patients with missing values to 1. We excluded patients with missing information on dispensed drugs after having performed a sensitivity analysis, as knowledge of opioid intake is needed for the PAR Risk Score. The sensitivity analysis on the 94 excluded patients was performed by setting the opioid variable for all these patients to either 0 or 1.

## Statistical analyses

Patient characteristics of the cohort were reported as number and frequencies. We calculated the raw PAR-Risk Score values for each patient as the sum of the predictor variables present at time of discharge (S1 Table). The raw PAR-Risk Score values were used in a univariable logistic regression to predict PARs within 30 days, as labelled as PAR case by the SQLape algorithm. We calculated C statistics, Brier score, and the 'le Cessie—van Houwelingen—Copas—Hosmer unweighted sum of squares test for global goodness of fit' as performance statistics of the univariable logistic regression. Additionally, we visualised the calibration by plotting the observed proportion at risk per PAR Risk Score point versus the predicted risk for each point weighted by the respective number of participants.

To compare the influence of a single predictor variable on the outcome to those of the original study, we analysed the unadjusted association between the single predictor variable and the outcome PAR using univariable logistic regression. Analogously, a multivariable logistic regression was performed to assess the independent association between individual predictors and the outcome [18].

We additionally categorised the patients into the three risk groups: *low*, *medium*, and *high risk* based on the raw PAR-Risk Score values using the original threshold levels of <3, 3–10, and >10, respectively. This grouping was redone with adapted threshold levels, which were re-calculated by grouping the patients into tertiles based on the raw PAR-Risk Score values of our patients, analogously to the development phase. Using the original as well as the adapted threshold levels, we calculated the observed proportion at risk, the predicted risk (S4 Table), and odds ratios (ORs) for PAR. We calculated sensitivity, specificity, positive predictive values, and negative predictive values with which the model classifies patients into the different risk groups by comparing each group to the *low* risk group.

All analyses were performed in R 3.6.1 [19] with the additional packages: tidyverse [20], lubridate [21], rms [22], pROC [23], and caret [24]. The *p*-values calculated in this report assume a significance level of .05.

## Ethics approval

The Swiss ethics committee approved the protocol for the study for which the data were originally extracted (EKNZ Project ID: 2018–01000). The committee also approved the amendment for the study presented here. The data were extracted anonymously, informed patient consent was not required.

## Results

Out of 8,252 patients hospitalised between December 2016 to November 2018, we included 5,985 patients in our study by applying the defined exclusion criteria (Fig 2). Of the eligible patients, 340 patients (5.7%) were identified as having experienced a PAR by the SQLape software, whereas it was 562 (7.7%) in the derivation patient cohort [12]. Patient characteristics are depicted in Table 1. The mean age of the patients was 79.7 (± 7.7) years with a mean of 16.9 (± 7.8) number of drugs of and a mean length of stay of 8.8 (± 6.5) days. The frequency of each predictor of PAR and non-PAR patients is shown in S1 Fig. Due to missing data, we set the PAR-Risk Score variable hyperkalaemia to 0 for 432 patients (7.2%). The 94 (1.5%) patients who were excluded due to missing data on dispensed drugs showed an underrepresentation of PAR cases with just one PAR case.

## Model specification

The distribution of the raw PAR-Risk Score values across SQLape-defined PAR versus non-PAR patients is presented in the supplement (S2 Fig). Unadjusted and adjusted associations

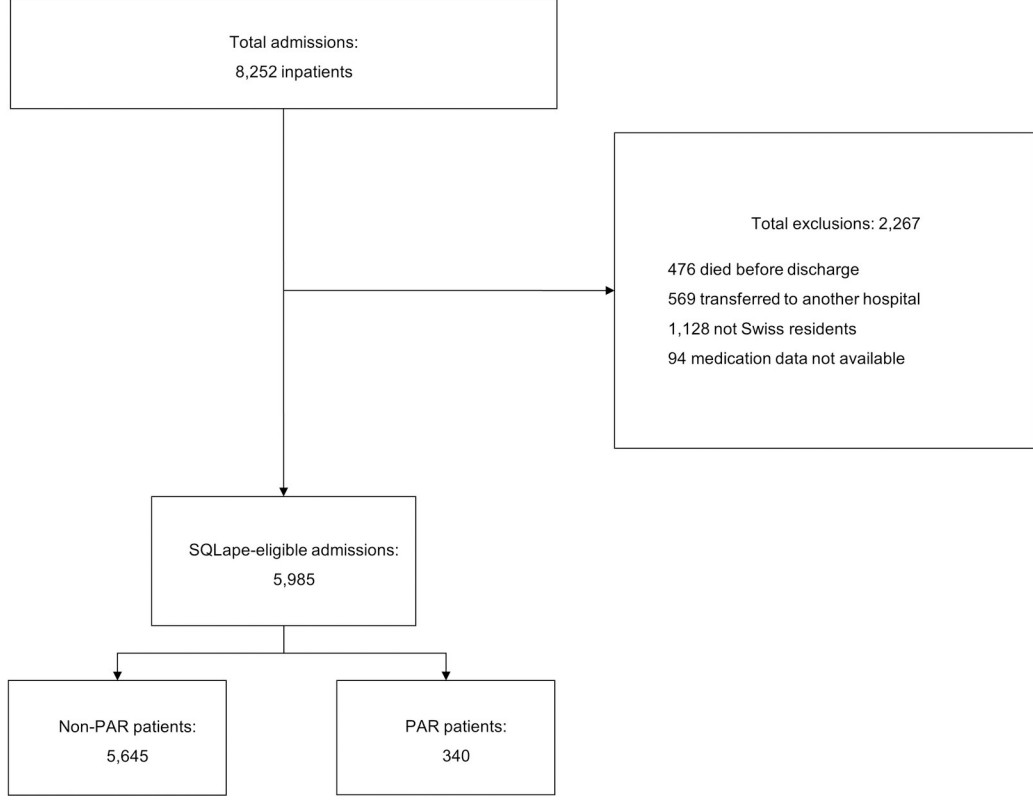

**Fig 2. Dataset generation with applied exclusion criteria.**

**Table 1. Patient characteristics of the cohort.**

| Patient characteristics at hospital discharge | n | % | Blanc *et al*. n (%) |
|---|---|---|---|
| Total number of patients | 5,985 | - | 7,317 (-) |
| SQLape defined PAR cases | 340 | 5.7 | 562 (7.7) |
| Age | | | |
| 65–75 years | 4,071 | 68.0 | 1,555 (21.3) |
| ≥76 years | 1,914 | 32.0 | 2,896 (39.6) |
| Male sex | 2,808 | 46.9 | 3,993 (54.6) |
| Length of hospital stay: | | | |
| ≤4 days | 1,570 | 26.2 | 2,259 (30.9) |
| >4 days | 4,415 | 73.8 | 5,058 (69.1) |
| Admission in previous 6 months | 1,360 | 22.7 | 2,041 (27.9) |
| Opioids* | 1,589 | 26.5 | 1,795 (24.5) |
| Number of drugs dispensed | | | |
| <5 | 115 | 1.9 | 1,671 (22.8) |
| 6 to 10 | 4,687 | 78.3 | 2,469 (33.7) |
| >10 | 1,183 | 19.8 | 3,177 (43.3) |
| Hyperkalaemia ($K^+$ >5.5 mmol/L)* | 16 | 0.3 | 685 (9) |
| Comorbidity | | | |
| Acute myocardial infarction | 275 | 4.6 | 1,048 (14.3) |
| Acute respiratory disease | 981 | 16.4 | 1,260 (17.2) |
| AIDS | 0 | 0.0 | 25 (0.3) |
| Anaemia | 1,235 | 20.6 | 2,138 (29.2) |
| Arrhythmia | 2,131 | 35.6 | 1,342 (18.3) |
| Cancer | 629 | 10.5 | 762 (10.4) |
| Metastatic carcinoma | 408 | 6.8 | 280 (3.8) |
| Cerebrovascular disease | 917 | 15.3 | 268 (3.7) |
| COPD/asthma | 776 | 13.0 | 1,043 (14.3) |
| Chronic ischemic heart disease | 1,512 | 25.3 | 497 (6.8) |
| Cognitive troubles/dementia | 741 | 12.4 | 201 (2.8) |
| Connective tissue disease | 90 | 1.5 | 64 (0.9) |
| Diabetes with organ damage | 467 | 7.8 | 152 (2.1) |
| Gastrointestinal ulcer | 137 | 2.3 | 100 (1.4) |
| Hepatic cirrhosis | 69 | 1.2 | 276 (3.8) |
| Heart failure | 1,361 | 22.7 | 1,314 (18.0) |
| Hypertension | 4,165 | 69.6 | 1,723 (23.6) |
| Infectious disease (except pneumonia and sepsis) | 1,733 | 29.0 | 1,655 (22.6) |
| Intoxication or adverse drug reactions | 87 | 1.5 | 918 (12.6) |
| Mental and behavioural disorders due to alcohol | 238 | 4.0 | 639 (8.7) |
| Paraplegia/hemiplegia | 197 | 3.3 | 84 (1.2) |
| Peripheral vascular disease | 543 | 9.1 | 186 (2–5) |
| Pneumonia | 619 | 10.3 | 1,353 (18.5) |
| Renal failure | 1,461 | 24.4 | 1,678 (22.9) |
| Sepsis | 189 | 3.2 | 542 (7.4) |

Notes: SQLape = Striving for Quality Level and Analyzing of Patient Expenses software [11]; PAR = Potentially avoidable readmissions; AIDS = Acquired immune deficiency syndrome; COPD = Chronic obstructive pulmonary disease.

* = variables with missing values.

**Table 2. Unadjusted and adjusted associations between predictor and potentially avoidable readmission (PAR).**

| Predictor | Non-PAR (n = 5645) | | PAR (n = 340) | | Univariable analysis | Multivariable analysis |
|---|---|---|---|---|---|---|
| | n | % | n | % | OR (95%-CI)* | |
| Admission in previous 6 months | 1,254 | 22.2 | 106 | 31.2 | 1.59 (1.25–2.01) | 1.39 (1.08–1.77) |
| Length of hospital stay | 4,146 | 73.4 | 269 | 79.1 | 1.37 (1.05–1.8) | 1.08 (0.82–1.44) |
| Anaemia | 1,134 | 20.1 | 101 | 29.7 | 1.68 (1.32–2.13) | 1.45 (1.12–1.85) |
| Heart failure | 1,256 | 22.2 | 105 | 30.9 | 1.56 (1.23–1.98) | 1.41 (1.09–1.81) |
| Hypertension | 3,919 | 69.4 | 246 | 72.4 | 1.15 (0.91–1.48) | 1.1 (0.86–1.42) |
| Acute myocardial infarction | 259 | 4.6 | 16 | 4.7 | 1.03 (0.59–1.67) | 0.95 (0.53–1.59) |
| Chronic ischemic heart disease | 1,420 | 25.2 | 92 | 27.1 | 1.1 (0.86–1.41) | 1.04 (0.8–1.35) |
| Diabetes with organ damage | 432 | 7.7 | 35 | 10.3 | 1.38 (0.95–1.96) | 1.18 (0.8–1.69) |
| Cancer | 584 | 10.3 | 45 | 13.2 | 1.32 (0.94–1.81) | 1.08 (0.71–1.59) |
| Metastatic carcinoma | 376 | 6.7 | 32 | 9.4 | 1.46 (0.98–2.09) | 1.23 (0.76–1.94) |
| Opioids | 1,466 | 26.0 | 123 | 36.2 | 1.62 (1.28–2.03) | 1.4 (1.1–1.78) |
| Hyperkalaemia | 14 | 0.2 | 2 | 0.6 | 2.38 (0.37–8.56) | 1.75 (0.27–6.44) |

Notes

* = The low risk group was used as reference for comparison with the medium and high risk groups. OR = Odds ratio; CI = Confidence interval.

between each predictor variable and the outcome are provided in Table 2. Out of the 12 predictor variables only five were found to be significantly associated with PAR, as identified by SQLape: previous admission within six months, length of hospital stay, anaemia, heart failure, and opioids. Multivariable logistic regression identified the four predictor variables previous admission within six months, anaemia, heart failure, and opioids to be significantly associated with the outcome. Based on the tertiles of our patients, the adapted threshold levels for the three risk categories *low*, *medium*, and *high* risk were PAR-Risk Score values of <12, 12 to 25, and >25, respectively.

## Model performance

The results of the univariable logistic regression of the raw PAR-Risk Score value on 30-days PAR is presented in the supplement (S2 Table). The univariable regression analysis yielded a C statistic of 0.605 (95% CI 0.575–0.635). The graph of the receiver-operating curve is provided in the supplement (S3 Fig). The Brier score was 0.053, indicating decent accuracy. The calibration plot indicated a lack of fit (Fig 3), which was also supported by the goodness-of-fit test with a *p*-value of <0.01. A summary of the goodness-of-fit test statistic is provided in the supplement (S3 Table).

The sensitivity analysis on patients with missing serum potassium levels showed only small changes in the C statistic: 0.600 (95% CI: 0.570–0.633) by setting hyperkalaemia to one. The sensitivity analysis on the 94 excluded patients with missing content on dispensed medication showed only small changes in the C statistic as well: 0.608 (95% CI: 0.579–0.638) by setting opioids to zero, and 0.607 (95% CI 0.577–0.637) by setting opioids to one.

Table 3 shows the frequency of PAR and non-PAR patients and the association between the odds of having a PAR given the threshold categories. When comparing *medium* and *high* risk groups to *low* risk groups, patients were at higher odds of being hospitalised for the adapted threshold levels. For the original threshold levels, only the high risk group was significantly at higher odds of being hositalised.

Performance measures for original and adapted thresholds are presented in Table 4. Using the original thresholds, the model classified patients into the *high* risk group with a sensitivity

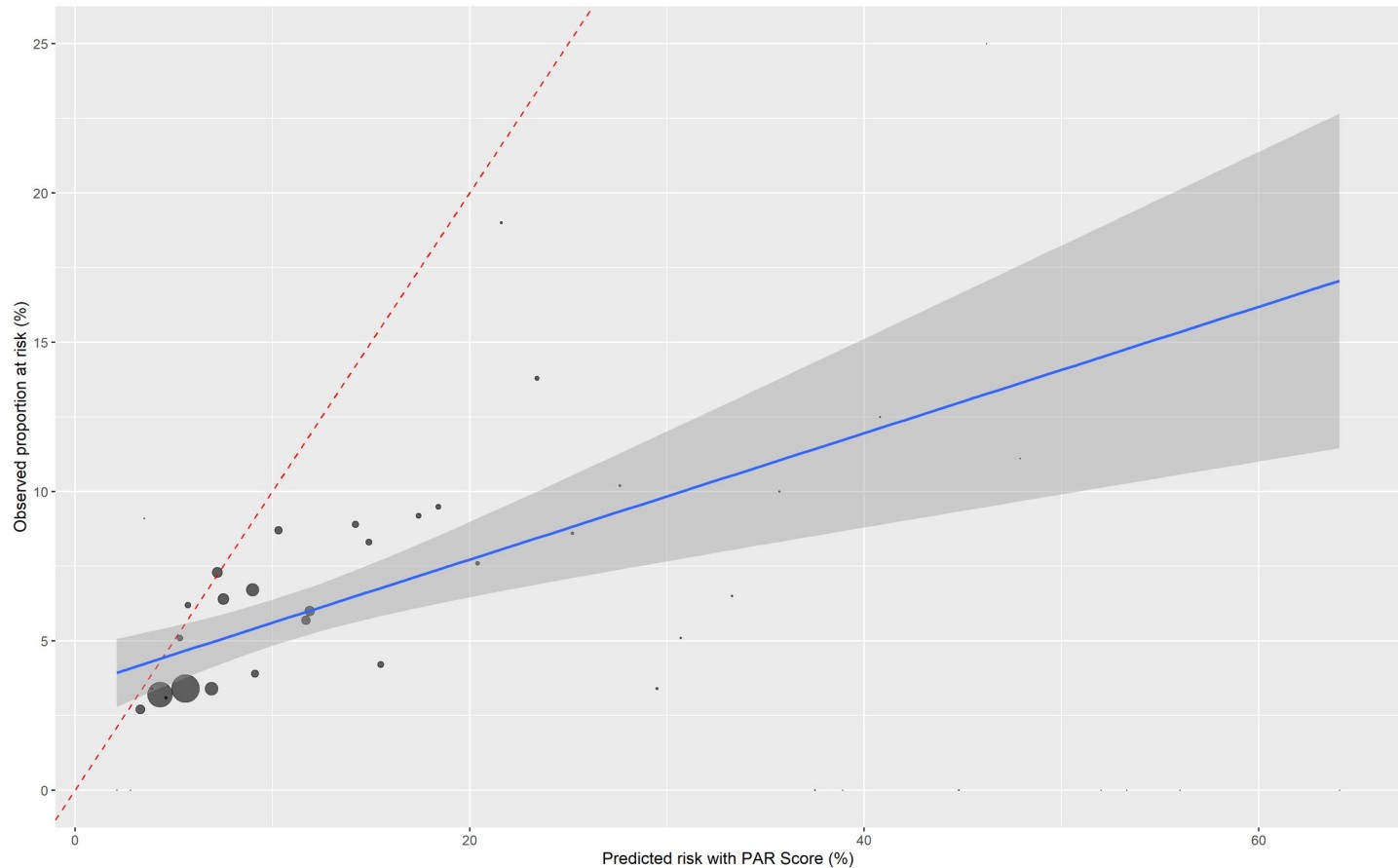

**Fig 3. Calibration plot of the PAR Risk Score weighted by number of participants.**

of 95.8% and a specificity of 10.4% when comparing to the *low* risk groups. Re-calculating the thresholds generally improved specificity (*high* to *low*: 53.4%) whilst reducing sensitivity (*high* to *low*: 67.9%). The predicted probabilities for each risk group using the original and the new threshold levels are presented in Table 5. For comparison, the observed proportion at risk and predicted probabilities in the derivation and validation cohort of the original publication are

**Table 3. Contingency table and odds ratios describing the association between risk group and potentially avoidable readmission (PAR).**

| Threshold category | Thresholds | PAR (n = 340) | Non-PAR (n = 5645) | OR (95%-CI)* |
|---|---|---|---|---|
| Original threshold | | | | |
| Low | <3 | 9 | 293 | 1 |
| Medium | 3–10 | 127 | 2826 | 1.46 (0.74–2.91) |
| High | >10 | 204 | 2526 | 2.63 (1.33–5.18) |
| Adapted threshold | | | | |
| Low | <12 | 72 | 2044 | 1 |
| Medium | 12–25 | 116 | 1818 | 1.81 (1.34–2.45) |
| High | >25 | 152 | 1783 | 2.42 (1.82–3.23) |

Notes

*The low risk group was used as reference for comparison with the medium and high risk groups. OR = Odds ratio; CI = Confidence interval.

**Table 4. Performance measures with which the model classifies patients into the different risk groups.**

|  | *Low* vs *medium* | *Low* vs *high* |
|---|---|---|
| Original threshold |  |  |
| Sensitivity (%) | 93.4 | 95.8 |
| Specificity (%) | 9.4 | 10.4 |
| Positive Predictive Value (%) | 4.3 | 7.5 |
| Negative Predictive Value (%) | 97.0 | 97.0 |
| Adapted threshold |  |  |
| Sensitivity (%) | 61.7 | 67.9 |
| Specificity (%) | 52.9 | 53.4 |
| Positive Predictive Value (%) | 6.0 | 7.9 |
| Negative Predictive Value (%) | 96.6 | 96.6 |

presented as well. The predicted patients at risk was overpredicted, yet comparable to the observed values for the *low* and *medium* risk groups, but twice as high for the *high* risk group (predicted: 15.9%, observed: 7.5%).

## Discussion

The aim of this study was to externally validate the PAR-Risk Score using retrospective data from a Swiss tertiary teaching hospital. Within this dataset, 340 patients (5.7%) were labelled as having experienced a PAR by the SQLape algorithm. For this dataset, we calculated the C statistic of the PAR-Risk Score to be 0.605, which was reported to be higher for the internal validation (0.687) [12]. The difference in the Brier score, indicating better accuracy in our dataset (Brier score: 0.053 versus 0.064), may be biased by the low number of patients with the outcome PAR [25]. The four predictor variables admission in previous 6 months, anaemia, heart failure, and opioids, showed significant associations with PAR in the multivariable analysis. Of these, anaemia, heart failure, and opioids showed a stronger association in our dataset than in the one used for the internal validation. When using the PAR-Risk Score to categorise the patients into three risk groups (*low*, *medium*, *high*) according to the original thresholds (<3,

**Table 5. Observed versus predicted risk for potentially avoidable readmission (PAR).**

|  | Risk group | Observed proportion at risk (%) | Mean predicted risk (%) |
|---|---|---|---|
| External validation |  |  |  |
| Original threshold | Low | 3.0 | 3.3 |
|  | Medium | 4.3 | 5.7 |
|  | High | 7.5 | 15.9 |
| Adapted threshold | Low | 3.4 | 4.7 |
|  | Medium | 6.0 | 8.0 |
|  | High | 7.9 | 18.6 |
| Original publication [12] |  |  |  |
| Derivation cohort | Low | 2.6 | 3.1 |
|  | Medium | 5.2 | 5.0 |
|  | High | 12.9 | 13.1 |
| Validation cohort | Low | 2.4 | 3.5 |
|  | Medium | 5.7 | 5.7 |
|  | High | 13.1 | 13.9 |

3–10, and >10), patients in the *high* risk group were at statistically significant higher odds (2.63, 95% CI 1.33–5.18) of being readmitted within 30 days compared to *low* risk patients.

We identified that having an admission in previous 6 months, anaemia, heart failure, and opioids as predictor variables from the PAR-Risk Score that had a significant association with PAR in our dataset. These variables are also part of other hospital readmission models [26–28]. Administrative variables such as previous admission to the hospital and length of stay were found to be frequently associated with hospital readmission [7,8]. Length of stay was not significantly associated with PAR in our study, but this result may have been influenced by dichotomising the variable (i.e., length of stay longer than four days or not). Heart failure as predictor variable has been extensively studied and is often reported as a risk factor for readmission [5,7,8,27], which was again confirmed for our cohort of patients. In contrary, anaemia is less frequently reported as risk factor for readmission, but included as predictor in some studies [7,29]. Anaemia has been shown to be associated with mortality in patients with chronic heart failure [30]. Opioid use is associated with medication-related harm in elderly patients, is a well-known cause for adverse drug events, and is linked to readmission [26,31,32]. This was true for our cohort of patients as well.

Variables included in readmission risk prediction generally include some mix of medical comorbidity data and prior use of medical services [8], with the final model depending on local characteristics and dataset limitations. Illness severity, overall health and functioning, and social determinants of health are frequently disregarded [8], with Herrin *et al.* showing that readmissions occurring after seven days are associated with non-hospital factors such as geodemographic characteristics and community-related factors [33]. Illness severity, overall health and functioning, and social determinants may be poorly accessible from administrative hospital data. Chin *et al.* were able to demonstrate that the reason for readmission due to hospital-level quality rapidly declined within the first seven days after discharge, meaning that readmissions after this time period are more susceptible to geodemographic characteristics reflecting social and community-related factors. This puts the 30-day readmission rate as outcome into question [34]. In Switzerland, a readmission of 18 days is important for the hospital's economical management. Therefore, prediction models considering non-hospital factors and reducing the readmission definition of 30 to 18 days might be more meaningful, and clinically and economically useful.

We assume that overall prediction accuracy would improve when accounting for the low incidence of PARs in the dataset during the model development stage. With the original thresholds, the model achieved a sensitivity of 95.8% and a specificity of just 10.4% to classify patients into the *high* rather than the *low* risk group. Adapting the thresholds influenced this ratio but did not markedly affect positive and negative predictive values. In regression-based prediction models, an imbalanced dataset will bias the prediction, leading to high accuracy for the majority group, while the minority group will show poor accuracy [35]. Possible techniques to account for class imbalance are basic resampling techniques (e.g., random over-/undersampling) or more sophisticated techniques such as the synthetic minority oversampling technique (SMOTE) [36].

The HOSPITAL score is another risk prediction model distinguishing *low*, *intermediate*, and *high* risk groups for 30-days PAR, which was derived from 10,731 discharges [13]. The model includes the predictive variables haemoglobin (<12 g/dL), discharge from an oncology service, sodium level (< 135 mEql/L), procedure during the index admission (≥1), index type of admission (emergency admission as opposed to elective), number of admissions during the last 12 months (0–1, 2–5, >5), and length of stay (≥5 days). In its external validation, comprising 117,056 patients from 9 hospitals and 4 countries, the HOSPITAL score showed a discriminatory power of 0.72 (C statistic, 95% CI 0.72–0.72) and a Brier score of 0.08 [14]. In the data

subset of Switzerland, comprising 8,971 patients with 524 SQLape-identified PARs (5.8%), the score showed a C statistic of 0.68.

## Limitations

Our external validation was conducted in older patients than in the development study, and had a required hospitalisation stay longer than 48 hours. This was a limitation of the available data and could have potentially influenced the model's performance. However, the main risk group for rehospitalisation are older patients [37], and predictive models need to perform for these patients. Based on clinical considerations, we decided against using imputation techniques for missing potassium values and information on dispensed drugs. If irregularities of potassium levels were to be expected, appropriate lab work would have been ordered. To investigate the impact of our considerations, we performed a sensitivity analysis, only showing small changes in the C statistic. No documented medication dispensing is an unlikely scenario for hospitalised patients over 65 years of age, suggesting incorrect data entry. We, hence, decided on excluding these 94 patients. Sensitivity analyses again confirmed the low impact of this decision on the model's performance. Another necessary deviation from the development phase was the unavailability of haemoglobin values in our dataset, which limited the definition of anaemia to the concerned ICD-codes [38]. This may have resulted in an underestimation of the prevalence of anaemia in our cohort as compared to the development phase (20.6% vs. 29.2%). Presence of anaemia, however, only assigns 2 points to the total PAR-Risk Score. We believe that adaptations of published prediction models to the local characteristics are frequently necessary. An additional limitation is that SQLape only identifies patients with unplanned readmissions to the same hospital. Hence, there is a possibility for misclassifying patients that went to another hospital for their readmission.

## Strengths

The strengths of this study stem from the decision to attempt a replication of the development for the model's external validation whilst simultaneously investigating the individual predictor variables rather than just the finished model. Staying as close to the development of the model as possible enabled statements about its validity and generalisability. Investigating the individual and combined impact of the predictor variables allowed for insights into the robustness of the predictors for future prediction models on PAR. This study additionally highlights the importance of externally validating prediction models by using a dataset derived from the cohort for which the model is intended to be applied to.

## Conclusion

In this external validation of the PAR-Risk Score in an internal medicine patient cohort of a 360-bed hospital and a mean age of 79.7 years, the model's overall performance was limited. Whilst higher scores were associated with an increased risk for PAR and patients in the high-risk group showed a statistically significant association with a 30-day readmission, the achieved C statistic as measure of discriminatory power was poor. This study confirms previous admission, length of stay, heart failure, and opioid use as potentially generalisable predictor variables of PAR. This study additionally displays the necessity of repeating the validation of published models with a local dataset prior to their use.

## Supporting information

**S1 Fig. Frequency of predictors of PAR vs. non-PAR patients.**
(DOCX)

**S2 Fig. Comparison of the distribution of raw PAR-Risk Score values in non-PAR and PAR group.** The dashed line indicates the original threshold levels (<3, 3–10, >10). The dot-dashed line indicates the adapted threshold levels (<12, 12–25, >25).
(DOCX)

**S3 Fig. Receiver operating curve of the univariable logistic regression.** C-statistic = 0.605.
(DOCX)

**S1 Table. Predictors and points for the calculation of the raw PAR-Risk Score.**
(DOCX)

**S2 Table. Results of the univariable logistic regression using the raw PAR-Risk Score values to predict PAR by SQLape.**
(DOCX)

**S3 Table. Goodness of fit test statistic of the univariable logistic regression.**
(DOCX)

**S4 Table. Coefficients of the multivariable regression of the original study.** The predicted risk was calculated by applying the scoring of the original study to each patient and then calculating the mean predicted risk by group.
(DOCX)

**S1 Data.**
(CSV)

## Acknowledgments

We would like to thank Melanie Berger for extracting the ICD-10 codes and explaining the rules of administrative hospital coding. Additionally, we also thank Leo Steinberger for extracting the data from the hospital's clinical data warehouse.

## Author Contributions

**Conceptualization:** Lukas Higi, Angela Lisibach, Dominik Stämpfli.

**Data curation:** Angela Lisibach.

**Formal analysis:** Lukas Higi, Angela Lisibach.

**Funding acquisition:** Monika Lutters, Andrea M. Burden.

**Methodology:** Lukas Higi, Angela Lisibach, Patrick E. Beeler, Anne-Laure Blanc, Dominik Stämpfli.

**Project administration:** Dominik Stämpfli.

**Supervision:** Monika Lutters, Andrea M. Burden.

**Writing – original draft:** Lukas Higi, Angela Lisibach, Dominik Stämpfli.

**Writing – review & editing:** Patrick E. Beeler, Monika Lutters, Anne-Laure Blanc, Andrea M. Burden.

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
