## [Decision Letter · Decision Letter 0]

18 Aug 2021

PONE-D-21-23537

External validation of the PAR-Risk Score to assess hospital readmission risk in internal medicine patients

PLOS ONE

Dear Dr. Stämpfli,

Thank you for submitting your manuscript to PLOS ONE. After careful consideration, we feel that it has merit but does not fully meet PLOS ONE’s publication criteria as it currently stands. Therefore, we invite you to submit a revised version of the manuscript that addresses the points raised during the review process.

We look forward to receiving your revised manuscript.

Kind regards,

Gianluigi Savarese

Academic Editor

PLOS ONE

Journal Requirements:

2. In ethics statement in the manuscript and in the online submission form, please provide additional information about the patient records/samples used in your retrospective study. Specifically, please ensure that you have discussed whether all data/samples were fully anonymized before you accessed them and/or whether the IRB or ethics committee waived the requirement for informed consent. If patients provided informed written consent to have data/samples from their medical records used in research, please include this information.

3. PLOS ONE has specific requirements for studies that are presenting a new method or tool as the primary focus, including a newly developed or modified questionnaire or scale (https://journals.plos.org/plosone/s/submission-guidelines#loc-methods-software-databases-and-tools.) One requirement is that the questionnaire or scale must be openly available under a license no more restrictive than CC BY. In light of this, before we proceed, please include a copy of your questionnaire or scale as a Supporting Information file (in the original language) or provide a link if it is available through an online repository.

Reviewers' comments:

Reviewer's Responses to Questions

**Comments to the Author**

1. Is the manuscript technically sound, and do the data support the conclusions?

Reviewer #1: Partly

Reviewer #2: Yes

2. Has the statistical analysis been performed appropriately and rigorously? 

Reviewer #1: Yes

Reviewer #2: Yes

3. Have the authors made all data underlying the findings in their manuscript fully available?

Reviewer #1: Yes

Reviewer #2: Yes

4. Is the manuscript presented in an intelligible fashion and written in standard English?

Reviewer #1: Yes

Reviewer #2: Yes

5. Review Comments to the Author

Reviewer #1: The authors of this manuscript have externally validated a risk score to assess hospital readmission risk in internal medicine patients. As is also clear from the results, it is extremely important to externally validate risk scores in different populations from where the risk score was created. The manuscript is well written, I have few questions that I hope the authors are able to address.

1. Wouldn't it have been an option to validate AND update the model to fit better with the new population? There has been an adaptation in the risk score cut offs but what about new variables that show better association and discrimination for a risk score?

2. As I understood cantonal is a Swiss healthcare system but might be difficult to understand for those that are not Swiss. Perhaps the authors could use secondary care as it is more widely understood?

3. The authors state that the derivation cohort was a primary care hospital but it is unclear to me what this means. Could the authors elaborate on this?

4. Why did the authors specify the age as above 65 years? 40% of the patients in the derivation cohort were under 65 years old? This was not an exclusion criteria in the initial study and could have potentially introduced bias and an underestimation of the c-statistic.

5. How much missing data was there? In the manuscript it is written that population mean was imputed, however this is an unrealiable form of imputing missing data as it shrinks standard errors and could alter the associations between variables. I would suggest to use multiple imputation or a Bayesian methods for single imputation.

6. How did the authors choose the new cut offs in the adapted risk score?

Reviewer #2: Dr. Higi and colleagues performed an external validation of the Potentially Avoidable Readmission Risk Score to evaluate the potentially avoidable hospital readmission risk in internal medicine patients from a Swiss cantonal hospital.

For that purpose, the authors used a 2-years cohort of internal medicine hospitalizations, longer than 48h and in patients older than 65, using the same algorithm (SQLape) of the derivation cohort, to identify the studied outcome (30-day potentially avoidable readmission – PAR). A total sample of 5985 patients was used considering data from electronic health records, but the authors had to deal with missing data.

It is expectable that the performance of prediction models would be poorer in the new sample than in the development population. However, those models should not be used spready before an established external validity. The work of the authors is of utmost relevance and this kind of analysis should be encouraged so the clinical impact of the previous manuscripts (development of scores) could (or not) be projected to the real-world practice. Still, some details could be enhanced in the reviewer opinion. Please find a detailed list below.

1. Title

a. I suggest adding “potentially avoidable” to the outcome “hospital readmission risk” to clearly represent the outcome that was measured.

2. Introduction

a. The medical context is well-explained but should be shortened.

3. Methods

a. The authors used correctly the TRIPOD check-list;

b. I suggest replacing “population” by “sample” or “participants”;

c. To consider an external validation the patients may be different from the derivation cohort. The main differences between the author’s sample and the development population should be emphasised explaining the reasons for that to constitute a sample for external validation. Also, the shared points: internal medicine patients;

d. The inclusion criteria (age > 65 and hospitalizations longer than 48 hours) were not clearly explained;

e. The age criteria could move the purpose of this external validation not only for the validation in another Centre within the same country where the PAR-Risk Score was derived (geographic validation, same country, other region), but also in a specific subgroup of patients, the elderly – these items should be incorporated at aims;

f. What were the main reasons for having missing values? Why did the authors use different approaches to deal with missing data in serum potassium and medication? Please clarify these topics.

g. A single imputation using sample mean was used to serum potassium levels in 514 (8.6%) patients. As the score uses a cut-off value of 5.5 mmol/L it is probably that all of those patients did not enter with a value which count to scoring 4 points, please discuss.

h. The authors performed missing imputation regarding medication and some sensitivity analyses considering two types of imputation and complete-cases analysis – this information is distributed between missing data and statistical analyses. Please report the type of missing imputation that was done within missing data subsection;

i. Please clarify lines 205-206, “the sensitivity analysis on the 94 excluded patients” or “the sensitivity analysis with/adding the 94 excluded patients”?

j. I’m not sure if any model updating was done or only the creation of new risk subgroups according to PAR-score distribution (please verify and complete 10e from TRIPOD accordingly).

4. Results

a. Please report how many PAR cases were reported on the 94 patients excluded due to missing data on dispensed drugs;

b. Please be cautious when reporting that 5.7% (PAR in the external cohort) is comparable with 7.5% (PAR in the derivation cohort) without any quantitative measure for comparison;

c. Table 1 – Please add the % of SQLape defined PAR cases in the derivation cohort;

d. Table 2 and Table 3 – Please add N by groups Non-PAR and PAR;

e. Figure 2 – Please note that the total exclusions (n=2267) do not correspond to the sum of the 4 exclusion criteria;

f. Figure S3 C-statistic (0.602) does not correspond to the C-statistic reported in the text (0.605) – line 202;

g. Table S3 – add observed value;

h. Regarding the risk stratification, medium and high-risk groups were at higher odds of having a PAR than low risk group considering the adapted threshold; while using the original threshold, only the high-risk group was significantly at a higher odd of having PAR versus low-risk category. Please reformulate lines 210-212;

i. Consider adding a calibration plot to table 5 to simplify reading and interpretation.

5. Limitations

a. The development study was not conducted in an older patient sample (~40% of patients were <65 years old) – please verify line 300;

b. Please consider including the limitations of the missing imputations methods. Would a multiple imputation provide more consistent results?

Other comments:

For PAR outcome and internal medicine patients’ population there are diverse prediction models. Therefore, an external validation of those prediction models on the same cohort can provide a comparison of the predictive performance between those models. Indeed, it is surprising that the HOSPITAL score reached similar discriminatory power and overall accuracy in the original/derivation cohort of the PAR-Risk score. It would be expectable that the new score to be superior since it was optimally designed to fit that data/sample. It would be interesting to check how the HOSPITAL score performs in this external cohort, as the medication seems to be the main difference between the PAR-Risk Score and HOSPITAL score and the missing information on medication data is one of the drawbacks of this external validation.

6. PLOS authors have the option to publish the peer review history of their article (what does this mean?). If published, this will include your full peer review and any attached files.

Reviewer #1: No

Reviewer #2: No

---

## [Author Response · Author response to Decision Letter 0]

1 Oct 2021

The authors would like to send their gratitude to the reviewers and the editor for their positive and constructive feedback. The comments and shared thoughts improved the overall work and added important details to the article.

To ease correspondence, we copied all comments into a table (uploaded as separate file). The reviewers’ comments are on the left side; the authors’ answers are on the right side. Indicated pages and lines refer to the clean version of the revised manuscript (track changes off).

We applied the following color scheme to our answers: Green states implementation, yellow states implementation which may be regarded as only partial, and orange states currently no implementation into the main body/only further clarification.

---

## [Decision Letter · Decision Letter 1]

28 Oct 2021

External validation of the PAR-Risk Score to assess potentially avoidable hospital readmission risk in internal medicine patients

PONE-D-21-23537R1

Dear Dr. Stämpfli,

We’re pleased to inform you that your manuscript has been judged scientifically suitable for publication and will be formally accepted for publication once it meets all outstanding technical requirements.

Kind regards,

Gianluigi Savarese

Academic Editor

PLOS ONE

---

## [Editor Report · Acceptance letter]

15 Nov 2021

PONE-D-21-23537R1 

External validation of the PAR-Risk Score to assess potentially avoidable hospital readmission risk in internal medicine patients 

Dear Dr. Stämpfli:

I'm pleased to inform you that your manuscript has been deemed suitable for publication in PLOS ONE. Congratulations! Your manuscript is now with our production department. 

Kind regards, 

on behalf of

Dr. Gianluigi Savarese 

Academic Editor

PLOS ONE